# Human-level Protein Localization with Convolutional Neural Networks

**Elisabeth Rumetshofer[1,2], Markus Hofmarcher[1,2], Clemens Röhrl[3], Sepp Hochreiter[1,2], Günter Klambauer[1,2]**

[1] LIT AI Lab, Johannes Kepler University Linz

[2] Institute for Machine Learning, Johannes Kepler University Linz

`{rumetshofer,hofmarcher,hochreit,klambauer}@ml.jku.at`

[3] Department of Medical Chemistry, Center for Pathobiochemistry and Genetics,
Medical University of Vienna

`clemens.roehrl@meduniwien.ac.at`

## Abstract

Localizing a specific protein in a human cell is essential for understanding cellular functions and biological processes of underlying diseases. A promising, low-cost, and time-efficient biotechnology for localizing proteins is high-throughput fluorescence microscopy imaging (HTI). This imaging technique stains the protein of interest in a cell with fluorescent antibodies and subsequently takes a microscopic image. Together with images of other stained proteins or cell organelles and the annotation by the Human Protein Atlas project, these images provide a rich source of information on the protein location which can be utilized by computational methods. It is yet unclear how precise such methods are and whether they can compete with human experts. We here focus on deep learning image analysis methods and, in particular, on Convolutional Neural Networks (CNNs) since they showed overwhelming success across different imaging tasks. We propose a novel CNN architecture "GapNet-PL" that has been designed to tackle the characteristics of HTI data and uses global averages of filters at different abstraction levels. We present the largest comparison of CNN architectures including GapNet-PL for protein localization in HTI images of human cells. GapNet-PL outperforms all other competing methods and reaches close to perfect localization in all 13 tasks with an average AUC of 98% and F1 score of 78%. On a separate test set the performance of GapNet-PL was compared with three human experts and 25 scholars. GapNet-PL achieved an accuracy of 91%, significantly ($p$-value $1.1e^{-6}$) outperforming the best human expert with an accuracy of 72%.[1]

## 1 Introduction

Proteins perform their function at specific times and locations within a cell. Understanding in which cell organelles, such as nucleus or mitochondria, a protein is located, is fundamental in understanding biological processes (Thul et al., 2017). The identification of mislocalized proteins may hint at cellular dysfunctions, advancing our knowledge about diseases. Microscopy-based methods assess changes in the sub-cellular localization or the abundance of proteins which helps to reveal their interactions in human cells.

Currently, the Human Protein Atlas project (HPA) is dedicated to annotating the location of all human proteins in a cell using a large battery of biotechnologies and approaches (Uhlen et al., 2010). A complementary and highly promising biotechnology that is used to localize proteins, is high-throughput fluorescence microscopy imaging (HTI), which is characterized by low costs and being time efficient (Pepperkok and Ellenberg, 2006). With this imaging technology, a selected protein can be stained with fluorescent antibodies and subsequently a microscopic image of the whole cell is taken. Together with the information of other stainings, such as the Hoechst staining of the cell nucleus, and the actin staining of the cytoskeleton, these images provide a rich source of

---

[1] Code and dataset are available at: `https://github.com/ml-jku/gapnet-pl`

information on the protein location (Stadler et al., 2013). Human experts are able to utilize those HTI images to localize proteins (Swamidoss et al., 2013). It is yet unclear how precise computational methods are and whether they achieve the performance level of human experts. High performance in localizing proteins is expected from deep learning methods since imaging data together with the annotation by the HPA project are an auspicious source of training data.

Convolutional Neural Networks (CNNs) are state-of-the-art for image analysis in a variety of fields (Krizhevsky et al., 2012), with variants proposed that increase performance significantly, e.g., DenseNet (Huang et al., 2017), ResNet (He et al., 2015), and Fully Convolutional Network (Long et al., 2015). Recently, CNNs have been successfully applied to analyze biological and medical images, in tasks such as detection of melanoma, performing on par with dermatologists (Esteva et al., 2017; Haenssle et al., 2018) or the automatic identification of malaria (Poostchi et al., 2018) based on brightfield microscopy images. Several authors have recently employed CNNs for high-throughput microscopy data (Dürr and Sick, 2016; Kraus et al., 2016; 2017; Pärnamaa and Parts, 2017; Godinez et al., 2017); however, certain characteristics of HTI data, such as relatively high resolution, prohibit application of standard models. As a result, the majority of the approaches focus on single-cell crops, i.e. segmented images containing a single cell, and thereby depend on data pre-processing steps. Or vice versa, (Liimatainen et al., 2018) have used semantic segmentation networks for localizing proteins and set the current-state-of-the art for this task to an F1-score of $0.51$. Overall, data pre-processing and cell segmentation algorithms still play a large role in HTI analysis.

With the big leap in performance caused by CNNs, comparisons of computational methods with humans or even human experts have recently appeared in literature. ResNet (He et al., 2015) has reached human level performance in image classification of general images. Esteva et al. (2017) have compared CNNs with expert performance at detecting melanoma in images of skin lesions. In the area of protein localization, Swamidoss et al. (2013) have conducted a comparison with two human experts. They worked with microscopy data on tissue level belonging to four classes. The performance of the machine learning approaches was still below the performance of the human experts. However, this pre-dates the rise in popularity of CNNs and therefore the authors extract features via several different methods and apply support vector machines and linear discriminant classifiers on top of these features. In this work, we aim at a more challenging task in which proteins have to be localized within 13 classes with multiple possible locations per sample.

Protein localization based on microscopy images poses a special problem in machine learning, namely, how to deal with weakly annotated data. The problem is that a set of instances, in this case cells, are labelled rather than single instances. All of those instances can provide hints about the correct classification. This is different from the problem setting of object recognition in MNIST, CIFAR, and ImageNet, where typically an image clearly represents a class. Current methods ameliorate this problem either by learning a high-level representation per image-patch or instance and then pooling over those patches, such as mean- or max-pooling, or noisy-and (Kraus et al., 2016) or even by joining the output layer predictions. Thus, these approaches collect hints from different patches. With GapNet-PL, we propose an alternative approach, in which hints are collected by global-average pooling layers. In contrast to other approaches, hints are not collected at a high-level representation but at the low-level convolutions. Thus, adaptions of our architecture could be an alternative to pooling representations or predictions for weakly annotated data.

The ideal approach to protein localization would be a generic, robust and fully automated pipeline that is as accurate or even more accurate than human experts. Even more desirable would be a platform to which HTI images could be uploaded and automatically annotated. In contrast, the current state-of-the-art in protein localization involves experts adjusting the segmentation algorithm, cell crop extraction and then applying Deep Learning on those with performance being supposedly below human experts. We offer a general approach that uses arbitrarily large input sizes, does not need segmentation but rather works on whole images and performs at the level of human experts or even above. With increased size and variability of training sets we suppose that the architecture could be used for localizing proteins in images from diverse biotechnological devices, cell lines and labs.

In the following, we apply current state-of-the-art methods as well as previous purpose-built approaches to high resolution images. We assess and compare their performance on the largest available public dataset of high resolution HTI data in the field of subcellular protein localization in

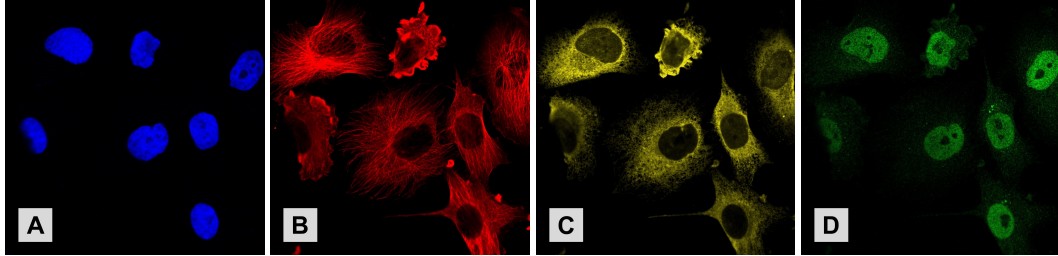

Figure 1: The four fluorescent channels of a sample from the dataset, namely (A) nucleus, (B) microtubules, (C) endoplasmic reticulum, (D) protein.

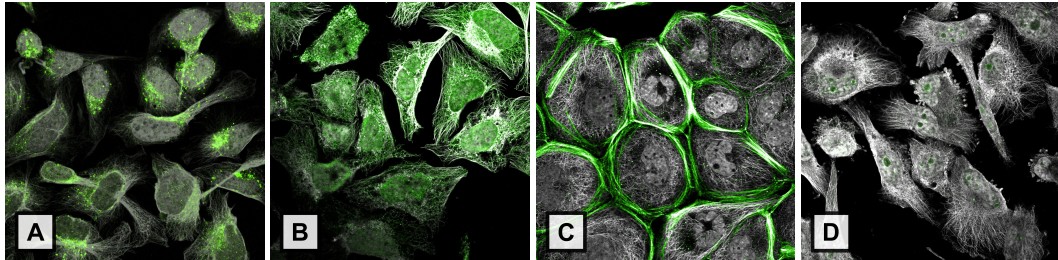

Figure 2: Exemplary samples from the dataset. Overlay of reference channels in gray and protein in green. The protein of interest is located in (A) golgi apparatus and vesicles, (B) cytosol, nucleus and plasma membrane, (C) actin filaments, (D) nucleoli and centrosome.

human cells. Furthermore, we compare the performance of human experts and scholars to the best performing network.

## 2 DATASET

All experiments were conducted on a dataset released for the "Cyto Challenge 2017" by the Congress of the International Society for Advancement of Cytometry (ISAC). The main challenge dataset contains 20,000 samples taken from the Cell Atlas (Thul et al., 2017) which is part of the Human Protein Atlas. The Cell Atlas contains images from antibody-based profiling using immunofluorescence confocal microscopy capturing multiple cells per image. These high resolution images of around 12,000 proteins across more than 20 cell lines are taken from different organs of the human body. Thus, the cell atlas is a rich source of cell images of high diversity with respect to shape, cell type, amount of cells and spatial relation of cells, which we use to perform our experiments on.

In the dataset, every sample consists of four high-resolution images corresponding to the different fluorescent channels. The channels represent three reference channels for different stained subcellular structures (nucleus, endoplasmic reticulum and microtubules) and one channel with the stained protein of interest (see Figure 1 and 2). For each sample, the task is to determine in which of the 13 major organelles the protein of interest appears, where multiple organelles are possible. Thus, each sample is associated with a binary label vector indicating the presence or absence of the protein in each of these 13 organelles. Most importantly, these labels are based on a consensus of several biotechnologies and human experts (Thul et al., 2017). In approximately 50% of the samples, the protein appears in more than one organelle and the distribution of proteins across organelles is slightly unbalanced (see Appendix Figure 5).

The dataset was provided as a collection of high resolution TIFF-files for every channel. Samples with at least one corrupted channel were removed which reduced the dataset from 20,000 to 18,756 samples. Due to differences in the capturing process channel-images are encoded differently, e.g., in different color channels and with varying dimensions (1728x1728, 2048x2048, 3072x3072) and were converted to grayscale if necessary. In a subsequent step these were normalized to zero mean and unit variance. To keep the quality of the original images and not lose finegrained details from

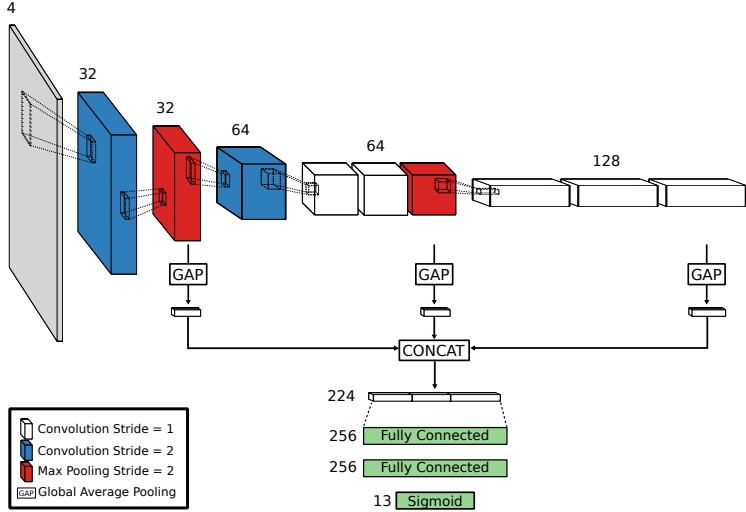

Figure 3: GapNet-PL architecture

interpolation, samples were not resized but kept at their original resolution. The pre-processed samples were then randomly split into a training (80%), a validation (10%) and a test (10%) set. For the comparison with human experts and scholars a subset of 200 examples, in which the protein appears in only one organelle ("multi-class" rather than "multi-task"), was chosen from the test set (see Appendix Figure 6).

## 3 METHODS

**GapNet-PL**  We propose an architecture designed specifically to process high-throughput microscopy images. Our architecture, which we call GapNet-PL, can deal with high resolution images and therefore has the possibility to learn from fine structures within images as they do not have to be downscaled. We achieve this via a two-step approach. In a first step we use an encoder consisting of several convolutional layers, some with a stride of two, interspersed with max-pooling layers to learn abstract features on different spatial resolutions.

In the second step, we reduce the feature maps from three different layers via *global average pooling* to a size of one pixel and concatenate the resulting feature vectors. This pooling operation can also mitigate the influence of weak labels. The resulting features, representing different spatial resolutions, are then passed on to a fully connected network with two hidden layers for the final prediction. This architecture has a relatively low number of parameters (in our case around 600,000) and can be tailored to a specific task very easily, e.g., by adding more convolutional blocks in the encoder and passing more features to the second stage. As a side effect, this architecture can process inputs of arbitrary spatial resolution on inference.

For efficient training and especially to reduce memory consumption, which is a problem specific to high-resolution images, we use the SELU activation function (Klambauer et al., 2017) instead of Rectified Linear Unit (ReLU) with Batch normalization. As a result, our architecture can be trained on high-resolution images with very short training times when compared to other architectures while achieving new state-of-the-art performance on our task. For our experiments we use an initial learning rate of 0.01, dropout in the fully connected layers of 30% and a batch size of 40.

**DenseNet**  Densely Connected Convolutional Networks (Huang et al., 2017) are currently among the best architectures for a variety of image processing tasks. The basic idea of DenseNet is to re-use features learned on early layers of a network, containing fine-grained localized information, on higher layers which have a more abstract representation of the input. This is achieved by passing feature maps of a layer to all consecutive layers (within certain boundaries). A stated benefit of this architecture is that it does not have to re-learn features several times throughout the network. Hence, the individual convolutional layers have a relatively small number of learned filters.

The authors present different variations of the architecture. Due to memory restrictions the smallest variant, DenseNet-121 (see Appendix Table 4), was chosen for the comparison in this work. The reduction rate of transition layers is set to 0.5 and the growth rate is $k = 32$. As proposed, Batch normalization and ReLU activation function are applied before every convolutional layer. For training a batch size of 3 and an initial learning rate of 0.1 was determined.

**Convolutional Multiple Instance Learning (Convolutional MIL)**   In Kraus et al. (2016) the authors introduce a CNN designed specifically for HTI data with a focus on the problem of weak labels, i.e. that microscopy images not only contain cells of the target or labeled class but also cells that do not comply with the label. The authors propose to tackle this problem with *multiple instance learning* (MIL), where cells belonging to the class label of an image are identified automatically while the influence of other cells on the result of the model is down-weighted by using a special pooling function called *noisyAND*.

This function follows the assumption that a sample belongs to a class if the number of cells of this class exceeds a threshold which is learned for each class individually. The authors implement their model using a fully convolutional approach (FCN) allowing them to train on full images with weak labels and apply this model to images of single-cell crops. Furthermore, the pooling function introduces an additional hyper-parameter $a$ controlling the slope of the function. Suggested values for $a$ in Kraus et al. (2016) are 5, 7.5 and 10. We use $a = 5$ for our final model after a comprehensive hyper-parameter search showed this to be the best value for our data. We train the model with an initial learning rate of 0.01 and a batch size of 32.

**Multi-scale Convolutional Neural Network (M-CNN)**   The M-CNN model (Godinez et al., 2017) is designed for phenotype classification of human HTI data and was benchmarked on 8 separate datasets, half of them dealing with protein localization for different cell types (HeLa, CHO, A-531 and MCF-7). The main idea of the architecture is to combine features extracted from the input at several spatial resolutions. This is achieved by scaling the original image dimensions to $width/s$ and $height/s$, where $s \in \{1, 2, 4, 8, 16, 32, 64\}$. These scaled versions of the input are processed by three convolutional layers and the feature maps of the last layer, respectively, are downscaled via pooling to the smallest resolution. Then, the feature maps are concatenated and combined via 1x1 convolution and passed on to a fully connected layer and the output layer. For this model we determined a batch size of 22 and an initial learning rate of 0.01.

**DeepLoc**   DeepLoc (Kraus et al., 2017) was designed for subcellular protein localization in yeast cells. The network consists of 8 convolutional layers in total, with max pooling layers after the second, fourth and last layer. The proposed model was trained on small crops of size 64x64 centered on x and y coordinates of cells extracted via CellProfiler (Carpenter et al., 2006) from larger images. Since the model was designed for considerably smaller inputs we had to adapt the architecture to make training possible (see Appendix Table 5 for detailed changes to the original architecture). Furthermore, we replaced ReLU with Batch normalization with SELU since it performed better in our experiments. The model is trained with batch size of 2 and an initial learning rate of 0.001.

**FCN-Seg**   Liimatainen et al. (2018) used a fully-convolutional network architecture to learn segmentation maps of the input images. Each channel is first processed by separate network-paths with three convolutional layers followed by max-pooling with an increasing number of units. The output of these paths are then combined by element-wise summation. This combination is processed with one convolutional layer followed by the final convolutional layer with 13 filters which provides a low-resolution segmentation of the inputs. These segmentation maps are binarized by a threshold to obtain the final classification, where the threshold for each class is optimized on the training set. As this method is designed for the dataset at hand, we followed the training procedure of the authors and use hyper-parameters as stated in the paper. Furthermore, we use the same framework and use default values for hyper-parameters not mentioned in the paper, such as learning rate.

## 3.1 MODEL TRAINING AND EVALUATION

To conduct a fair comparison we re-implemented all methods and optimized the most relevant hyper-parameters for each method. Since not all methods are designed for large inputs, a comparison on full resolution images is not feasible due to memory restrictions. However, the nature of our data

imposes limitations on small crops. First, very large or elongated cells are often not captured in full. Second, the problem of weak labels gets amplified, as a label might not conform to the whole crop, whereas on high resolution the label at least corresponds to parts of the image. Experiments with different crop sizes corroborate this (see Appendix Figure 8). Furthermore, Liimatainen et al. (2018) also trained on images containing multiple cells with relatively high resolution.

We train all models on 1024x1024 pixel crops, where one random crop per training sample is extracted in every epoch. With the exception of DeepLoc, every method in our comparison is designed for processing such large inputs. For DeepLoc, we also tested smaller input sizes of 72x72 and 224x224 pixels and subsequent averaging the predictions of those smaller crops, which performed worse (see Appendix Figure 8). For validation and testing the whole image is used by cropping several 1024x1024 pixel patches covering the whole image. A final prediction for a sample is derived by applying average pooling over the predictions of all patches. All methods consider this problem as a multi-task setting of 13 binary classification tasks. Therefore, all networks contain an output layer with 13 units, sigmoid activations and cross entropy loss.

The batch size for the models depends on their memory consumption and is chosen as large as possible to fit on a typical GPU with 11GB memory (NVIDIA GTX 1080 Ti). As an optimizer we use Stochastic Gradient Descent (SGD) with momentum of 0.9. To stabilize training we use gradient clipping where the gradients are normalized if their global L2 norm exceeds a certain threshold (which is set to 5). For optimal results we use a learning rate schedule, specifically the learning rate is halved when model performance plateaus. The initial learning rate is determined via a hyperparameter search on the validation set. To avoid overfitting the following regularization techniques are applied: L1 norm of $1e^{-7}$, L2 norm of $1e^{-5}$. Dropout is used as proposed in the respective publications. All models are trained until convergence and the checkpoint with best performance on the validation set is used for inference on the test set.

### 3.2 COMPARISON WITH HUMAN EXPERTS AND SCHOLARS

To determine the practical applicability of our model, we conducted a comparison of the GapNet-PL performance with a) 3 human experts in pathobiology who frequently work with fluorescence microscopy images and b) 25 scholars, i.e. graduate and undergraduate students with a life science background that were given special training. For this comparison, we consider only samples where the protein binds to a single location within cells in order to simplify the task and decrease workload for the participants. We fine-tuned the last layer of our model, which was trained in a multi-task setting, on a multi-class subset (11,848 samples) of the training set.

In a first interactive session with one of the experts the presentation of samples was decided. For each sample participants were presented with four images, each of the three channels overlayed with the protein, and the protein channel alone (see Appendix Figure 9). All experts were given the opportunity to investigate the training set *ad libitum*. They stated that they would be capable to localize proteins with the provided data. The setup for scholars differed in that they received a 90 minute training session where training samples for each class with its characteristics, as defined by experts, were presented. The training set was also provided to the scholars and they could browse it at will. All participants were provided with 200 samples from our held-out test set and had approximately one week to classify them. To simplify the annotation process we provided a web-based annotation tool.

## 4 RESULTS

The results of our method comparison are shown in Table 1. We found that the average performance across the 13 tasks of the compared methods ranges from an area under ROC (AUC) of 91% to 98%, with a mean AUC of 95%. Since for imbalanced learning scenarios the AUC may provide a too optimistic view of the models we also report the average performance of F1 score, precision and recall. The average F1 score over all tasks comprises a larger range from 47% to 78% with a mean of 65%. The table shows that the predictive performance of GapNet-PL, M-CNN and Dense-Net-121 are closer together and the remaining methods are behind by a large margin. We used a one-sided paired Wilcoxon test to test if our results are statistically significant. Overall, GapNet-PL significantly outperforms all compared methods.

| Method | F1 Score | $p$-value | Precision | Recall | AUC |
|---|---|---|---|---|---|
| GapNet-PL | **0.78 ± 0.09** | | 0.84 | **0.75** | **0.98 ± 0.02** |
| M-CNN | 0.75 ± 0.12 | 0.0232 | **0.90** | 0.66 | 0.97 ± 0.02 |
| DenseNet-121 | 0.73 ± 0.12 | 0.0434 | 0.75 | 0.74 | 0.97 ± 0.02 |
| DeepLoc | 0.52 ± 0.28 | 0.0009 | 0.79 | 0.45 | 0.91 ± 0.07 |
| FCN-Seg | 0.50 ± 0.18 | 0.0007 | 0.63 | 0.44 | 0.71 ± 0.08 |
| Convolutional MIL | 0.47 ± 0.31 | 0.0007 | 0.71 | 0.40 | 0.94 ± 0.06 |
| Liimatainen et al. (2018)[1] | 0.51 | - | 0.45 | 0.68 | - |

[1] performance estimate calculated on a different test dataset

Table 1: Performance comparison of different CNN architectures for protein localization. Liimatainen et al. (2018) state the winning results of the "Cyto Challenge 2017" on the unpublished official test set. For each method, the table reports the average F1 score and its standard deviation across 13 prediction tasks. The third column, $p$-value, reports the result of a one-sided paired Wilcoxon test with the null hypothesis that GapNet-PL and the respective method perform equally across prediction tasks. The consecutive columns report average precision, recall and area under ROC (AUC). GapNet-PL has outperformed all competing methods.

| Method | Accuracy | F1 Score | Precision | Recall |
|---|---|---|---|---|
| GapNet-PL | **0.91** | **0.82** | **0.75** | **0.95** |
| Expert 1 | 0.72 | 0.57 | 0.55 | 0.67 |
| Expert 2 | 0.66 | 0.49 | 0.44 | 0.64 |
| Expert 3 | 0.64 | 0.58 | 0.73 | 0.58 |
| Mean Scholars | 0.51 | 0.36 | 0.39 | 0.46 |
| Ensemble of experts | 0.71 | 0.60 | 0.60 | 0.68 |
| Ensemble of scholars | 0.71 | 0.57 | 0.60 | 0.65 |

Table 2: Predictive performance of GapNet-PL, 3 human experts and the group of scholars. The table shows the overall accuracy and average F1 score, precision and recall over 13 tasks. In each column, the best performing method is written bold. Additionally, results of ensembles of experts and scholars are given. An ensemble is calculated by majority vote. In case of ties, one of the tied classes was chosen randomly. We repeated this random sampling 1,000 times, such that the provided estimates for the ensembles are averages over those 1,000 samplings.

The performance of the compared methods also varies across the 13 localization tasks corresponding to cell organelles. GapNet-PL has the best performance on 9 tasks, M-CNN on 3 tasks and DenseNet-121 on 1 task. DeepLoc, FCN-Seg and Convolutional MIL are competitive for some tasks but show an inconsistent performance over all 13 tasks (see Appendix Table 7). Considering only the top 3 methods, on average the most challenging tasks are Centrosome, Intermediate filaments and Endoplasmic reticulum, while the easiest tasks for the 3 methods are Nucleus, Nuclear membrane and Mitochondria. We hypothesize that both the number of positive examples for each task, as well as, the complexity of the task itself influences the predictive performance (see Discussion Section 5).

| | Both Correct | GapNet-PL Correct | Human Expert Correct | Both Incorrect | $p$-value |
|---|---|---|---|---|---|
| Expert 1 | 133 | 48 | 10 | 9 | 1.1e-06 |
| Expert 2 | 124 | 57 | 8 | 11 | 2.6e-09 |
| Expert 3 | 121 | 60 | 7 | 12 | 2.1e-10 |

Table 3: Confusion matrix for comparison of GapNet-PL with human experts. $p$-values are calculated with a McNemar's test for equality of row and column marginal frequencies.

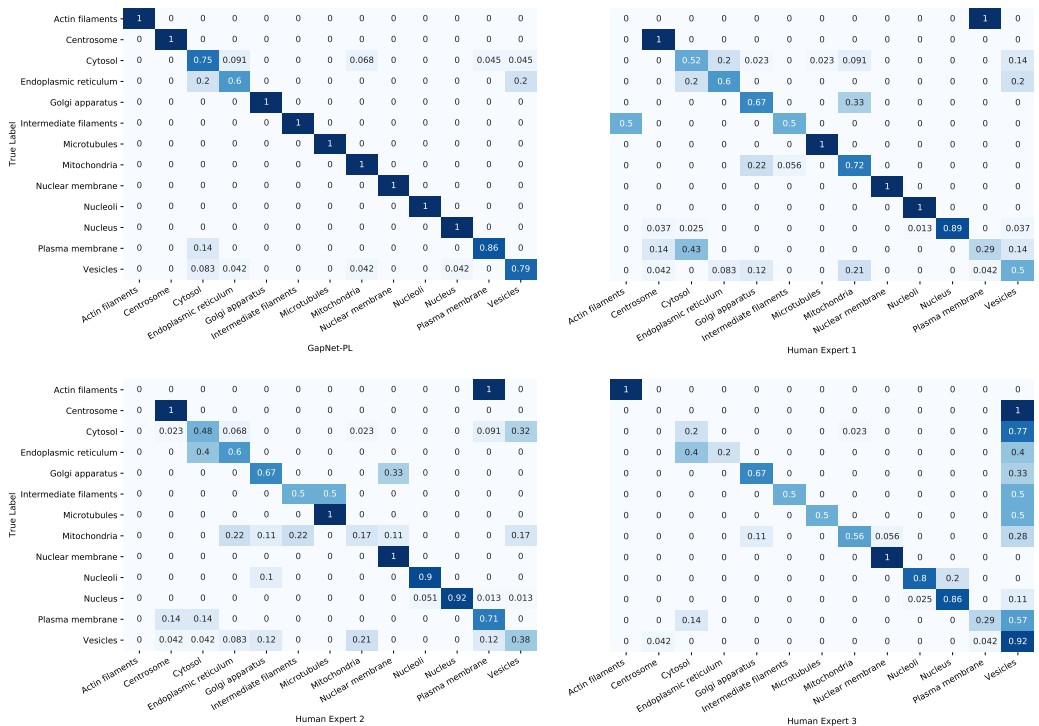

Figure 4: Confusion matrices for multi-class estimates of GapNet-PL (top left) and the 3 human experts. The CNN exhibits good performance across all classes, predicting 9 classes perfectly, while the human experts performance varies widely for several classes.

In our comparison of the best performing CNN against human experts and scholars, we found that the computational method could reach an accuracy of 91%. The human experts reached an accuracy of 64% to 72% (with an average of 67%) and scholars reached an average accuracy of 51% (see Table 2). To test if there is a significant difference between the performance of GapNet-PL and each of the human experts, we applied a McNemar test that resulted in $p$-values of $1.1e^{-6}$, $2.6e^{-9}$ and $2.1e^{-10}$ for human expert 1, expert 2, and expert 3, respectively. Hence, GapNet-PL has significantly outperformed all three human experts. We discuss potential biases this comparison could suffer from in Section 5. Table 3 shows the confusion matrices for the comparison of GapNet-PL with the human experts. In Figure 4 we show a more detailed break down on a per-task level. Evidently, several classes such as Nucleoli, Nuclear membrane and Nucleus are easily identifiable for experts while others, e.g. Plasma membrane and Cytosol, appeared to be challenging, resulting in large variability of predictions. In comparison, GapNet-PL shows a consistent performance and can predict most of the classes perfectly.

## 5 CONCLUSION AND DISCUSSION

In this work, we introduced a new neural network architecture *GapNet-PL* that was designed to overcome the challenges introduced by the nature of HTI data such as high resolution and weak labels. In a large study comparing convolutional neural networks for protein localization in human cells, GapNet-PL performs almost perfectly, achieving an average AUC of 98% and an F1 score of 78%, outperforming all compared computational methods while requiring less computational resources (see Appendix Table 6). Furthermore, we show that the best performing neural network significantly outperforms three human experts and ensembles of human scholars. Considering the individual tasks separately, a similar picture emerges. Our method is superior to the other methods in most of the tasks. M-CNN and DenseNet-121 show a competitive performance and win the remaining tasks. The overall performance of the top 3 methods is close to perfect in terms of AUC. However, the performance across the tasks varies, some of the classes are harder to learn (Cen-

trosome, Intermediate filaments and Endoplasmic reticulum), while Nucleus, Nuclear membrane and Mitochondria are quite easy tasks for the best performing CNNs. The relative frequency does not seem to be the only driver for this variability since the class "Nuclear membrane" is a strongly underrepresented class which can be predicted well with our tested methods.

We further investigated cases where GapNet-PL or human experts exhibit failures. Overall, there are 6 cases in the test set where both the human experts and GapNet-PL predict the wrong class and 18 samples misclassified by human experts but correctly by GapNet-PL. When both GapNet-PL and human experts misclassify a sample, the staining appears unspecific or the predicted classes are quite similar to the correct one (see Appendix Figure 10 and Figure 11). There are three classes that are frequently confused by both human experts and GapNet-PL: endoplasmatic reticulum, cytosol, and vesicles (see also Figure 4). In cases, where human experts fail, but GapNet-PL classifies correctly (see Appendix Figures 12-15), the staining can be variable across cells (Appendix Figure 12) or the staining is weak (Appendix Figure 13). For these difficult cases, GapNet-PL performs on average better than human experts. Based on our investigation of failures, we hypothesize that GapNet-PL is better for classes that have similar staining patterns, such as cytosol and vesicles, and in cases where there is some variability across cells, and when the staining is weak.

With respect to our comparison with the human experts, we are aware that our estimate of human performance at this task could suffer from potential biases such as (and not limited to) the following: a) Anchoring effect: The human experts investigated only a subset of the training data points and could be biased towards those; b) Confirmation bias: The human experts knew they would compete against an AI and probably thought the AI would be better, which could have decreased their motivation and thereby their performance; c) Response bias: There was interaction between human experts and the experimenter in an interactive session, such that the human experts knew that the experimenter's goal was to show that the AI can outperform humans and might have subconsciously supported this effort or – vice versa – may have motivated the human experts to perform even better than routinely; d) The human experts could also be performing better than usual since they knew they were in a testing situation; e) Limited sample size and selection bias: we only tested the performance of three human experts and the human experts were selected to be familiar with protein localization in microscopic images, but it is unclear how representative their performance is for possible human performance. It would be a tremendous effort to find a completely fair experimental setting, if this is possible at all, which is beyond the scope of this manuscript.

We have shown that GapNet-PL could be used to automatically annotate protein locations in immunostained HTI data and envision that it could be developed into a routine clinical application. It is a generic and robust method which can process input images of arbitrary size capable of learning images from various heterogeneous cell lines. It already exhibits a high predictive performance on a large and diverse dataset setting the state-of-the-art from $0.51$ to $0.78$ and could get even better with more data from different stainings and additional cell lines produced by other biotechnological devices from various labs.

## 6 ACKNOWLEDGMENTS

This project was supported by Bayer AG with Research Agreement 09/2017, Audi Electronic Venture GmbH with Research Agreement 12/2016 and LIT with grant LIT-2017-3-YOU-003. We thank the NVIDIA Corporation for the GPU donations, LIT with grant LSTM4Drive and the Audi JKU Deep Learning Center.

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

# 7 APPENDIX

## 7.1 DATASET

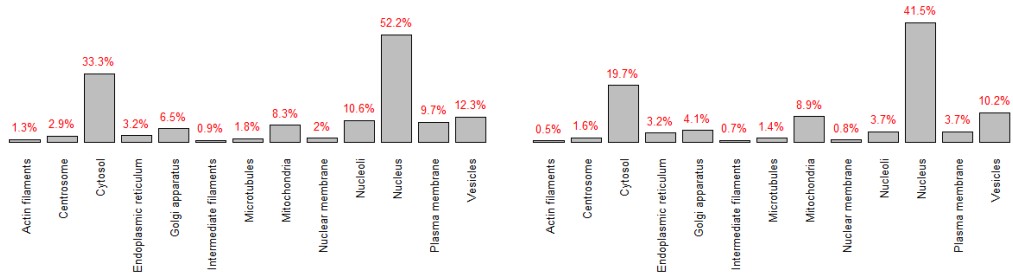

Figure 5: Class distributions for the whole dataset with 18,756 samples (left) and for all 11,484 multi-class samples (right).

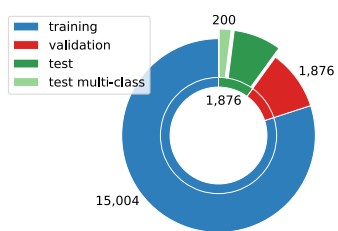

Figure 6: Visualization of the datasets. We used an 8:1:1 split of dataset into training (15,004 samples), validation (1,876 samples) and test (1,876 samples) sets. An additional test set was defined for the comparison with human experts and scholars, which is a subset of 200 samples from the test set with single-class labels.

## 7.2 ARCHITECTURES

### 7.2.1 DENSENET

| Layers | Output Size | DenseNet-121 | | DenseNet-169 | | DenseNet-201 | | DenseNet-264 | |
|---|---|---|---|---|---|---|---|---|---|
| Convolution | $112 \times 112$ | $7 \times 7$ conv, stride 2 | | | | | | | |
| Pooling | $56 \times 56$ | $3 \times 3$ max pool, stride 2 | | | | | | | |
| Dense Block (1) | $56 \times 56$ | $1 \times 1$ conv $3 \times 3$ conv | $\times 6$ | $1 \times 1$ conv $3 \times 3$ conv | $\times 6$ | $1 \times 1$ conv $3 \times 3$ conv | $\times 6$ | $1 \times 1$ conv $3 \times 3$ conv | $\times 6$ |
| Transition Layer (1) | $56 \times 56$ | $1 \times 1$ conv | | | | | | | |
| | $28 \times 28$ | $2 \times 2$ average pool, stride 2 | | | | | | | |
| Dense Block (2) | $28 \times 28$ | $1 \times 1$ conv $3 \times 3$ conv | $\times 12$ | $1 \times 1$ conv $3 \times 3$ conv | $\times 12$ | $1 \times 1$ conv $3 \times 3$ conv | $\times 12$ | $1 \times 1$ conv $3 \times 3$ conv | $\times 12$ |
| Transition Layer (2) | $28 \times 28$ | $1 \times 1$ conv | | | | | | | |
| | $14 \times 14$ | $2 \times 2$ average pool, stride 2 | | | | | | | |
| Dense Block (3) | $14 \times 14$ | $1 \times 1$ conv $3 \times 3$ conv | $\times 24$ | $1 \times 1$ conv $3 \times 3$ conv | $\times 32$ | $1 \times 1$ conv $3 \times 3$ conv | $\times 48$ | $1 \times 1$ conv $3 \times 3$ conv | $\times 64$ |
| Transition Layer (3) | $14 \times 14$ | $1 \times 1$ conv | | | | | | | |
| | $7 \times 7$ | $2 \times 2$ average pool, stride 2 | | | | | | | |
| Dense Block (4) | $7 \times 7$ | $1 \times 1$ conv $3 \times 3$ conv | $\times 16$ | $1 \times 1$ conv $3 \times 3$ conv | $\times 32$ | $1 \times 1$ conv $3 \times 3$ conv | $\times 32$ | $1 \times 1$ conv $3 \times 3$ conv | $\times 48$ |
| Classification | $1 \times 1$ | $7 \times 7$ global average pool | | | | | | | |
| Layer | | 1000D fully-connected, softmax | | | | | | | |

Table 4: DenseNet architecture variants (Huang et al., 2017). The authors present different variations of the architecture. Due to memory restrictions the smallest variant, DenseNet-121, was chosen for the comparison in this work. The reduction rate for the 1x1 convolution transition layer is set to 0.5 and the growth rate is k = 32.

### 7.2.2 DEEPLOC

| Layers | Convolution | Convolution | Max Pooling | Convolution | Convolution | Max Pooling | Convolution | Convolution | Convolution | Convolution | Max Pooling | Fully Connected | Fully Connected | Output |
|---|---|---|---|---|---|---|---|---|---|---|---|---|---|---|
| Units (original) | 64 | 64 | 64 | 128 | 128 | 128 | 256 | 256 | 256 | 256 | 256 | 512 | 512 | 19 |
| Units (adapted) | 32 | 32 | 32 | 64 | 64 | 64 | 96 | 96 | 96 | 96 | 96 | 128 | 128 | 13 |

Table 5: The architecture configuration taken from Kraus et al. (2017) and the adaptations for our comparison.

### 7.2.3 MULTI-SCALE CONVOLUTIONAL NEURAL NETWORK (M-CNN)

| | Scale factor s | Convolutional layer 1 | Convolutional layer 2 | Convolutional layer 3 | Pooling factor | Convolutional layer 4 | Pooling factor | Fully-connected layer | Output layer |
|---|---|---|---|---|---|---|---|---|---|
| **Input image** | 1 | 16 kernels 5 × 5 pixels | 16 kernels 5 × 5 pixels | 16 kernels 5 × 5 pixels | 64 | | | | |
| | 2 | 16 kernels 5 × 5 pixels | 16 kernels 5 × 5 pixels | 16 kernels 5 × 5 pixels | 32 | | | | |
| | 4 | 16 kernels 5 × 5 pixels | 16 kernels 5 × 5 pixels | 16 kernels 5 × 5 pixels | 16 | | | | |
| | 8 | 32 kernels 5 × 5 pixels | 32 kernels 5 × 5 pixels | 32 kernels 5 × 5 pixels | 8 | 1024 kernels 1 × 1 pixel | 2 | 512 units | $N_p$ units |
| | 16 | 32 kernels 5 × 5 pixels | 32 kernels 5 × 5 pixels | 32 kernels 5 × 5 pixels | 4 | | | | |
| | 32 | 32 kernels 5 × 5 pixels | 32 kernels 5 × 5 pixels | 32 kernels 5 × 5 pixels | 2 | | | | |
| | 64 | 64 kernels 5 × 5 pixels | 64 kernels 5 × 5 pixels | 64 kernels 5 × 5 pixels | 1 | | | | |

Figure 7: The architecture configuration taken from the appendix of Godinez et al. (2017).

### 7.3 MODEL TRAINING AND EVALUATION

| Method | Training Time/Epoch (sec) | Parameters |
|---|---|---|
| Convolutional MIL | 413 | 593,143 |
| GapNet-PL | 605 | 626,221 |
| M-CNN | 895 | 34,193,853 |
| DenseNet-121 | 2,176 | 6,970,317 |
| DeepLoc | 3,497 | 201,716,967 |

Table 6: Training times per epoch in seconds and number of parameters for all compared methods.

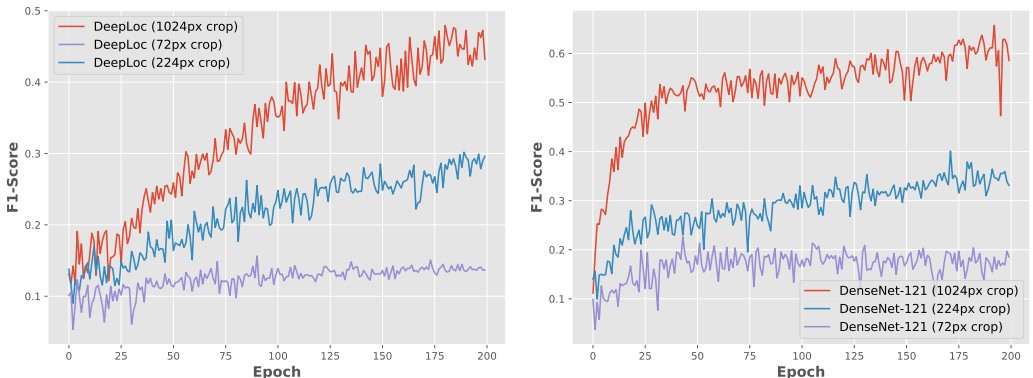

Figure 8: Comparison of different input sizes (72x72, 224x224 and 1024x1024 pixel) for DeepLoc and DenseNet on the validation set.

## 7.4 COMPARISON WITH HUMAN EXPERTS AND SCHOLARS

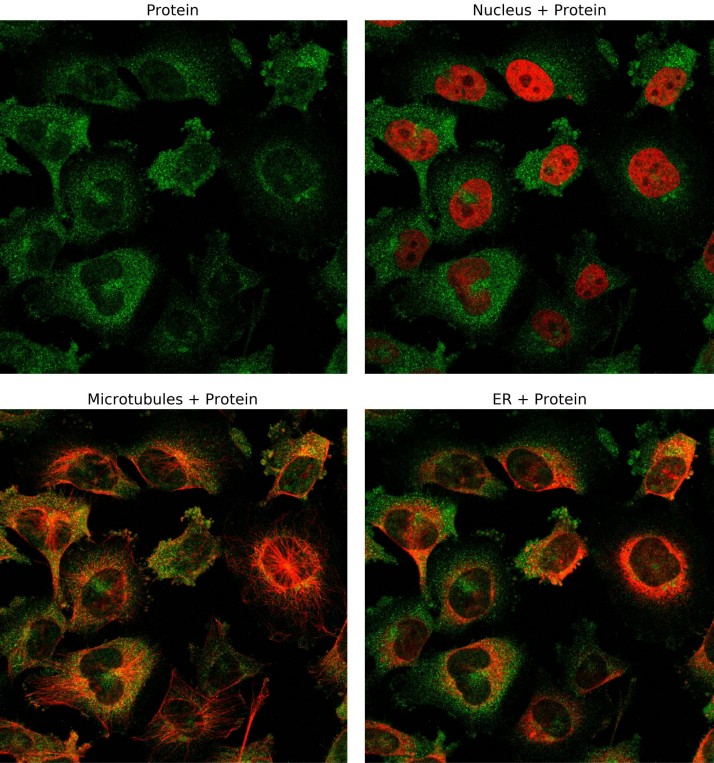

Figure 9: Visualization of samples provided to the human experts and scholars. Every sample was displayed in 4 separate images with the protein in green and the reference channels in red.

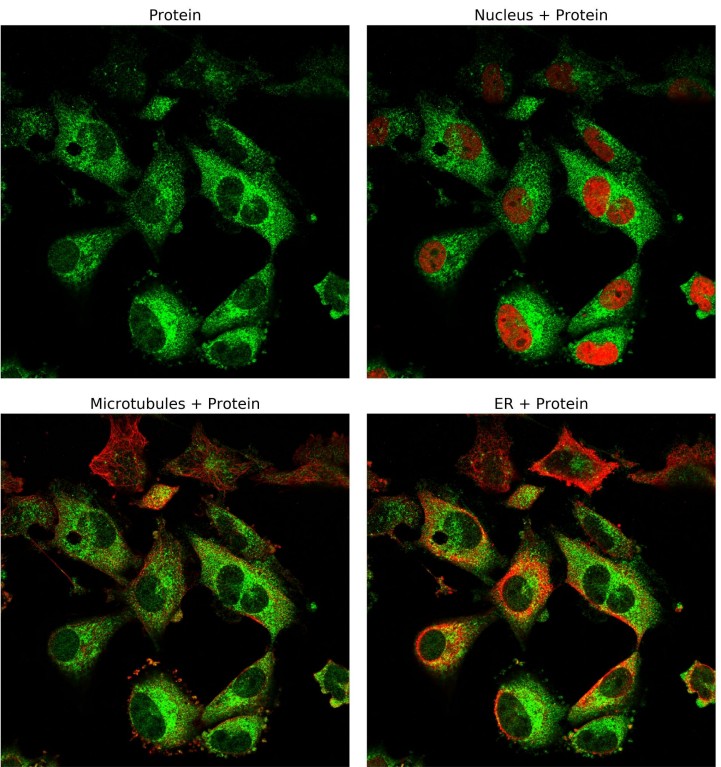

Figure 10: Sample misclassified by both GapNet-PL and all human experts. True class: Endoplasmic reticulum. GapNet-PL: Cytosol. Human experts: Cytosol, Cytosol and Vesicles. All three classes are similar structures outside the nucleus up to or near the boundaries of the cell.

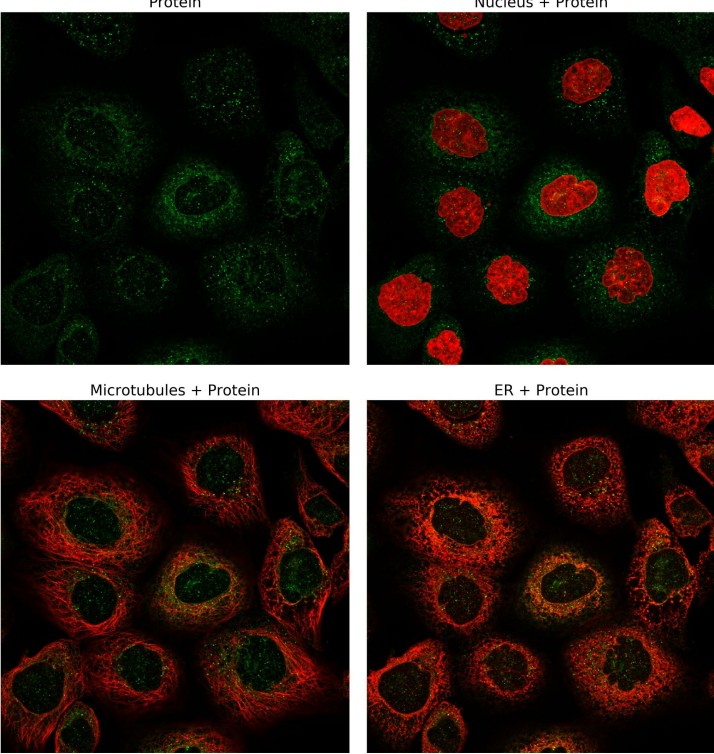

Figure 11: Sample misclassified by both GapNet-PL and all human experts. True class: Endoplasmic reticulum. GapNet-PL: Vesicles. Human experts: Vesicles, Cytosol and Vesicles. All three classes are similar structures outside the nucleus up to or near the boundaries of the cell.

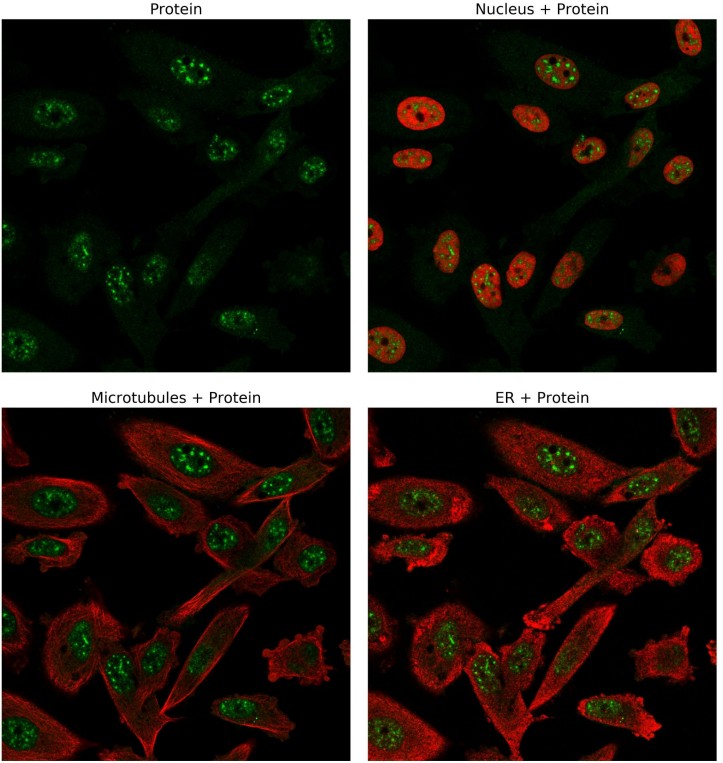

Figure 12: Sample misclassified by all human experts but correctly by GapNet-PL. True class: Nucleus. GapNet-PL: Nucleus. Human experts: Nucleoli, Nucleoli and Nucleoli.

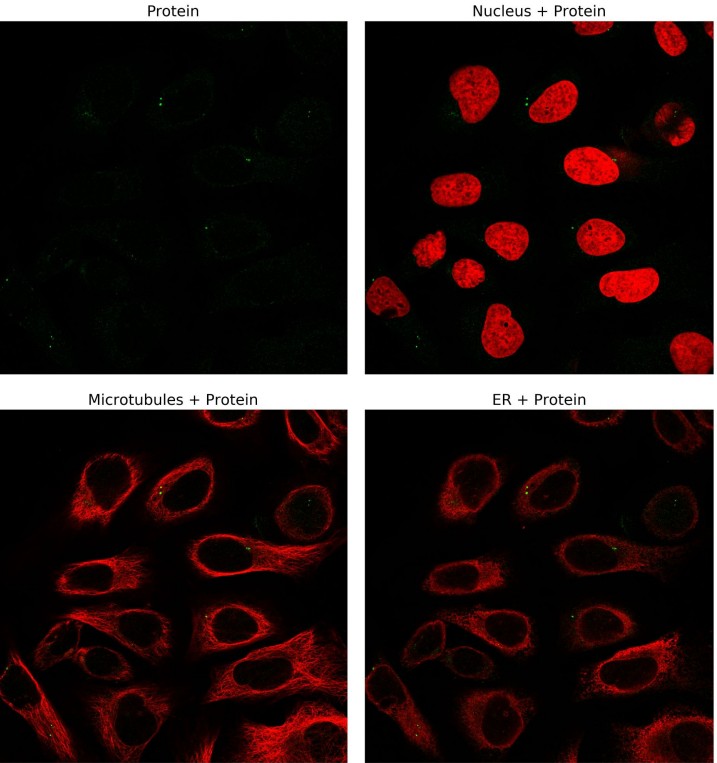

Figure 13: Sample misclassified by all human experts but correctly by GapNet-PL. True class: Vesicles. GapNet-PL: Vesicles. Human experts: Centrosome, Centrosome, Centrosome.

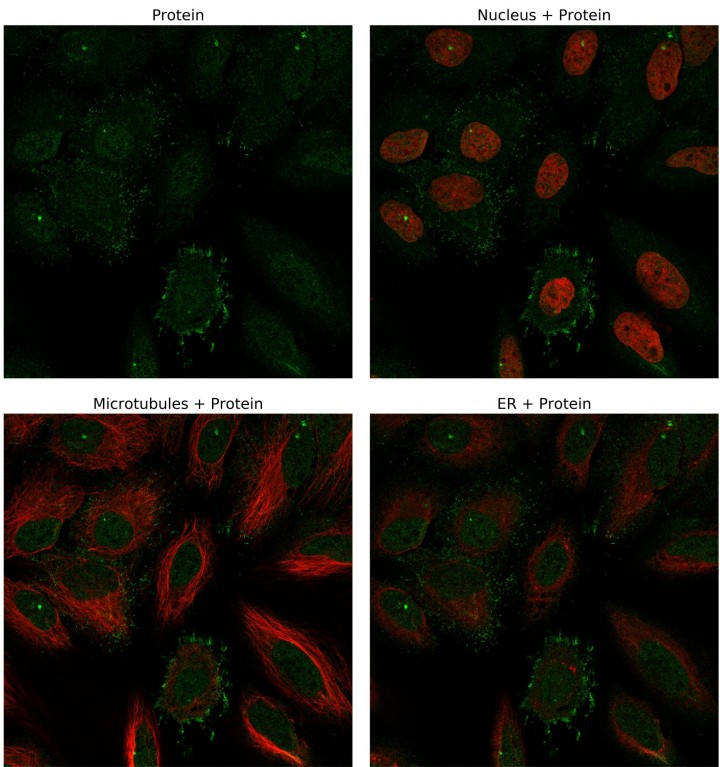

Figure 14: Sample misclassified by all human experts but correctly by GapNet-PL. True class: Plasma membrane. GapNet-PL: Plasma membrane. Human experts: Centrosome, Centrosome, Vesicles.

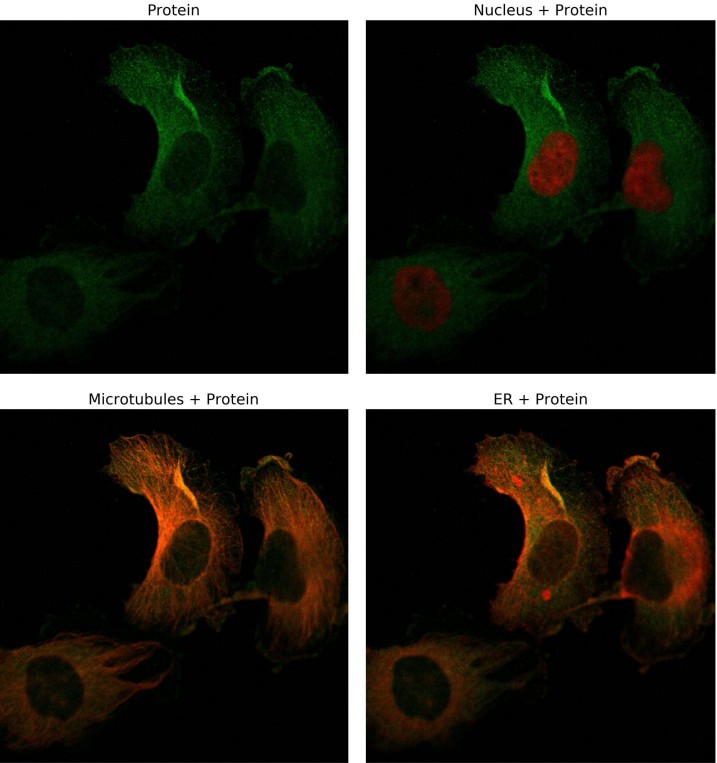

Figure 15: Sample misclassified by all human experts but correctly by GapNet-PL. True class: Cytosol. GapNet-PL: Cytosol. Human experts: Microtubul, Plasma membrane and Vesicles.

## 7.5 RESULTS

| Method | GapNet-PL | M-CNN | DenseNet-121 | DeepLoc | FCN-Seg | Convolutional MIL |
|---|---|---|---|---|---|---|
| Actin filaments | 0.682 | 0.651 | **0.719** | 0.125 | 0.341 | 0.000 |
| Centrosome | **0.659** | 0.522 | 0.598 | 0.039 | 0.299 | 0.000 |
| Cytosol | 0.776 | **0.805** | 0.789 | 0.784 | 0.701 | 0.735 |
| Endoplasmic reticulum | **0.672** | 0.629 | 0.645 | 0.538 | 0.301 | 0.338 |
| Golgi apparatus | **0.754** | 0.689 | 0.750 | 0.428 | 0.497 | 0.331 |
| Intermediate filaments | **0.800** | 0.625 | 0.490 | 0.000 | 0.222 | 0.000 |
| Microtubules | 0.767 | **0.901** | 0.853 | 0.696 | 0.438 | 0.712 |
| Mitochondria | **0.866** | 0.818 | 0.843 | 0.758 | 0.652 | 0.608 |
| Nuclear membrane | **0.922** | 0.907 | 0.756 | 0.675 | 0.553 | 0.692 |
| Nucleoli | **0.873** | 0.816 | 0.826 | 0.613 | 0.544 | 0.802 |
| Nucleus | **0.949** | 0.945 | 0.943 | 0.925 | 0.897 | 0.945 |
| Plasma membrane | **0.766** | 0.756 | 0.656 | 0.597 | 0.525 | 0.538 |
| Vesicles | 0.683 | **0.685** | 0.682 | 0.551 | 0.547 | 0.435 |
| Mean | **0.782** | 0.750 | 0.735 | 0.517 | 0.501 | 0.472 |

Table 7: F1 scores per task for each method. Overall, GapNet-PL performs best for 9 tasks, M-CNN for 3 tasks and DenseNet wins one task.

