# OpenReview forum: "Human-level Protein Localization with Convolutional Neural Networks"
_ICLR.cc/2019/Conference_

### Official Review · AnonReviewer3 · 2018-10-17
**A CNN that boosts the state of the art on an important image classification task in biology**

**Rating:** 8
**Confidence:** 4

**Review:**

This manuscript describes a deep convolutional neural network for
assigning proteins to subcellular compartments on the basis of
microscopy images.

Positive points:

- This is an important, well-studied problem.

- The results appear to improve significantly on the state of the art.

- The experimental comparison is quite extensive, including
  reimplementations of four, competing state-of-the-art methods, and
  lots of details about how the comparisons were carried out.

- The manuscript also includes a human-computer competition, which the
  computer soundly wins.

- The manuscript is written very clearly.

Concerns:

There is not much here in the way of new machine learning methods.
The authors describe a particular neural network architecture
("GapNet-PL") and show empirical evidence that it performs well on a
particular dataset.  No claims are made about the generalizability of
the particular model architecture used here to other datasets or other
tasks.

A significant concern is one that is common to much of the deep
learning literature these days, namely, that the manuscript fails to
separate model development from model validation. We are told only
about the final model that the authors propose here, with no
discussion of how the model was arrived at.  The concern here is that,
in all likelihood, the authors had to try various model topologies,
training strategies, etc., before settling on this particular setup.
If all of this was done on the same train/validation/test split, then
there is a risk of overfitting.

The dataset used here is not new; it was the basis for a competition
carried out previously.  It is therefore somewhat strange that the
authors chose to report only the results from their reimplementations
of competing methods.  There is a risk that the authors'
reimplementations involve some suboptimal choices, relative to the
methods used by the originators of those methods.

Another concern is the potential circularity of the labels.  At one
point, we are told that "Most importantly, these labels have not been
derived from the given microscopy images, but from other
biotechnologies such as microarrays or from literature."  However,
earlier we are told that the labels come from "a large battery of
biotechnologies and approaches, such as microarrays, confocal
microscopy, knowledge from literature, bioinformatics predictions and
additional experimental evidence, such as western blots, or small
interfering RNA knockdowns."  The concern is that, to the extent that
the labels are due to bioinformatics predictions, then we may simply
be learning to re-create some other image processing tool.

The manuscript contains a fair amount of biology jargon (western
blots, small interfering RNA knockdowns, antibodies, Hoechst staining,
etc.) that will not be understandable to a typical ICLR reader.

At the end, I think it would be instructive to show some examples
where the human expert and the network disagreed.

Minor:

p. 2: "automatic detection of malaria" -- from images of what?

p. 2: Put a semicolon before "however" and a comma after.

p. 2: Change "Linear Discriminant" to "linear discriminant." Also, remove
the abbreviations (SVM and LDA), since they are never used again in
this manuscript.

p. 5: Delete comma in "assumption, that."

p. 8: "nearly perfect" -> "nearly perfectly"

The confusion matrices in Figure 5 should not be row normalized --
just report raw counts.  Also, it would be better to order the classes
so that confusable ones are nearby in the list.

---

> ### Author Response · Authors · 2018-11-26
> **Reply to reviewer**
>
> We thank the reviewer for his/her positive comments. We have implemented and tested another method (of Liimatainen et al., 2018). Furthermore, we have put efforts in improving the human-computer competition by recruiting two additional experts to perform the task and 25 scholars, i.e. graduate undergraduate students with a life science background that were given special training. We found that the performance of experts ranges from an accuracy of 64% to 72% (average: 67%) and the performance of scholars ranges from an accuracy of to 34% to 66% (average: 51%).
>
> Concerns:
>
> 1.) Generalizability of the architecture
>
> We thank the reviewer for pointing out his concern. As communicated to the other reviewers, our architecture provides an alternative to ameliorate the problem of weakly labeled data. Instead of combining hints (pooling) of different instances at representation or prediction level, our network rather combines hints from low-level filter. We think that this problem is occurring in a number of different situations, such that researchers in other machine learning fields might find the GapNet-PL technique highly relevant to their problem, as well.
>
> 2.) Model development and re-implementation of competing methods
>
> We aimed at using the original implementations of the architectures and only made changes if the original architecture was impossible to run, which could happen if the original architecture was developed on smaller images. Otherwise, we only changed parameters if they led to an improvement of the validation metrics.
> The reviewer is correct that we only use a single train/val/test split, where all hyper-parameter optimization is done on the validation set in order to prevent overfitting. We also took care that the amount of searched hyper-parameters is similar for all methods (e.g. we tested two different initial learning rates for all competing methods).
> The GapNet architecture was developed on a different data set with similar characteristics (classification task of high-throughput fluorescence microscopy images) and then adapted to this dataset. We downsized the model slightly as the original task was more complex and then adjusted its most important hyper-parameters (such as learning rate) on the validation set. We will make this more explicit in the upcoming version of the paper, thanks for pointing this out.
>
> 3.) Circularity of labels
>
> We thank the reviewer for this important comment that led us to dive deeper into the actual annotation process of those images (Thul et al., 2017). It turned out that the description of the challenge data set was partly lacking with respect to the annotation process. We updated the respective information in the dataset section of the paper.
>
> 4.) Biology jargon
>
> We improved the introduction section and removed the biology jargon. Additionally, we now provide examples where humans and network disagree.
>
> Minor comments:
>
> Thanks, we corrected as suggested except for the last one. We decided to keep the normalized confusion matrices to be consistent with Esteva et al. (2017) and also because the visualization (coloring of the cells) is better.

---

### Official Review · AnonReviewer2 · 2018-11-02
**high-performance method with minor methodological contributions**

**Rating:** 5
**Confidence:** 3

**Review:**

The paper proposes a CNN variant tailored for high-resolution
immunofluorescence confocal microscopy data.  The authors show
that the method outperforms a human expert.

The proposed method is evaluated on benchmark instances
distributed by Cyto Challenge '17, which is presumably the best
data source for the target application.  Indeed, the method
performs better than several competitors plus a single human
expert.

The paper is well written and easy to follow.  I could not spot any
major technical issues.

This is an applicative paper targeting a problem that is very
relevant in bioinformatics, but it sports little methodological
innovation.  On the biological side, the contribution looks
significant.  Why not targeting a bioinformatics venue?


Detailed comments:

Papers that stretch multiple fields are always hard to review.  On
one hand, having contributions that cross different fields is a
high-risk (but potentially highly rewarding) route, and I applaud
the authors for taking the risk.  On the other hand, there's the risk
of having unbalanced contributions.

I think that the contribution here is mostly on the bioinformatics
side, not on the deep learning side.  Indeed, the method boils
down to a variant of CNNs.  I am skeptical that this is enough to
spark useful discussion with practitioners of deep learning
(although I could be wrong?).

Finally, I am always skeptical of "human-level" performance claims.
These are strong claims that are also hard to substantiate.  I don't
think that comparing to a *single* expert is quite enough.  The fact
that "the human expert stated that he would be capable to localize
proteins with the provided data" doesn't sound quite enough.  I
agree that the user study could be biased (and that "It would be
a tremendous effort to find a completely fair experimental
setting"), but, if this is the case, the argument that the method
reaches human-level performance is brittle.


Other remarks and questions:

- Why wasn't the dataset of Liimatainen et al. used for the
comparison?

- The authors say that "due to memory restrictions, the smallest
variant of DenseNet was used".  How much of an impact could have
this had on performance?

- "One random crop per training sample is extracted in every epoch".
Doesn't this potentially introduce labeling errors?  Did you observe
this to occur in practice?

- The authors claim that the method is close to perfect in terms
of AUC.  In decision-making applications, the AUC is a very
indirect measure of performance, because it is independent of
any decision threshold.  In other words, the AUC does not measure
the yes/no decisions suggested by the method.  Why is the AUC
important in the biological application at hand?  Why is it important
to the users (biologists, I suppose) of the system?

In particular, "our method performs nearly perfect, achieving an
average AUC of 98% and an F1 score of 78%" seems inconsistent
to me---the F1 is indeed "only" 78%.

- I would appreciate if there was a thorough discussion of the
failure mode of the expert.  What kind of errors did he/she
make?  How are these cases handled by the model?

---

> ### Author Response · Authors · 2018-11-26
> **Reply to reviewer**
>
> We thank the reviewer for his encouraging statement about our work.We believe that the results are also relevant for the machine learning community because our architecture tackles the problem of weakly labeled data different than other methods before.
>
> Ad relevance for deep learning
>
> Our contribution on the Deep Learning side is on the problem of dealing with weakly annotated data. Instead of single instances that are labeled, a set of instances is labeled. A machine learning method has to collect and combine hints from the instances for accurate predictions. Many hints across the image have to be collected in order to be able to correctly classify. This is rather more similar to sentiment detection than to object recognition (MNIST, CIFAR, ImageNet), in which typically a single instance has to be detected. We think that this problem is occurring in a number of different situations, such that researchers in other machine learning fields might find the GapNet-PL technique highly relevant to their problem, as well.
>
> Ad comparison with human experts
>
> As the reviewer was skeptical about comparisons with humans, so were the authors. Thus, we have taken efforts to improve the estimates of human performance and of expert performance: We recruited two more human experts to improve the performance estimate of human experts. We also tested 25 graduate and undergraduate students, we call scholars, with life science background and specific training at this task. Thus, we can now also provide a reliable estimate for knowledgeable, non-expert human performance.
> We adapted the respective paragraphs including these new results that provide a better view on human performance at this task.
>
> Other remarks and questions:
>
> 1.) Dataset of Liimatainen et al.
>
> They report the performance on the Challenge test set that has never been released. Therefore, we had to compare the methods on a different test set. However, we have now re-implemented the method of Liimatainen et al. and compared it on our test set, where it had a similar performance (F1 score of 0.50 on our test set and 0.51 on the challenge test set). Hence, all performance metrics are estimated on the same test set and are comparable.
>
> 2.) DenseNet
>
> Due to computational constraints, the computations for DenseNet were restricted to a single GPU on which we could only fit this variant. We hypothesize that there could be small improvement of performance with larger variants.
>
> 3.) Random crops
>
> In practice, we did not observe that an empty crop is propagated through the network, which empirically confirmed by testing around 100,000 random crops. However, we agree with the reviewer about the labeling problem: the data set can in general be considered as weakly annotated (see answer above) because the whole image and not the individual instances, i.e. cells, are labeled.
>
> 4.) AUC as performance measure
>
> The ranking is important in cases, where multiple proteins are tested and the research team wants to perform subsequent test of all proteins that are only located in, say, the nucleus. In this case they would start with the highest ranked protein and test the following proteins until they have found the one with the desired properties.
>
> 5.) Failure modes of human experts
>
> We now provide a list of images (see Appendix) which were frequently mis-labeled by the human experts and how the CNN handled those images including a thorough discussion.

---

### Official Review · AnonReviewer1 · 2018-11-02
**This is an application oriented paper with little technical contribution**

**Rating:** 4
**Confidence:** 4

**Review:**

This paper designed a GapNet-PL architecture and applied GapNet-PL, DenseNet, Multi-scale CNN etc. to the protein image (multi-labels) classification dataset.

Pros:

1. The proposed method has a good performance on the given task. Compared to the claimed baselines (Liimatainen et al. and human experts), the proposed architecture shows a much higher performance.

Cons:

1. The novelty of the proposed architecture is limited. The main contribution of this work is the application of CNN-based methods to the specific biological images.

2. The existing technical challenge of this task is not significant and the motivation of the proposed method could be hardly found in this paper.

3. The baselines are not convincing enough. Since the performance of Liimatainen et al. is calculated on a different test dataset, the results here are not comparable. The prediction from a human expert, which may vary from individuals, fails to provide a confident performance comparison.

4. Compared to the existing models (DenseNet, Multi-scale CNN etc.), the performance improvement of the proposed model is limited.

---

> ### Author Response · Authors · 2018-11-26
> **Reply to reviewer**
>
> We thank the reviewer for the assessment of our work.
>
> Ad Pros 1.)
>
> We thank the reviewer for this positive comment. We have even improved our baseline estimates a lot.
>
> Ad Cons 1.)
>
> We agree with the reviewer that this is an application paper (such as references [1-3]) - for which ICLR specifically called. However, the problem posed by these images is rather general than specific (see point 2).
> We improved the introduction section to emphasize the main contribution of our work, which we see in the development of the architecture and that it treats the problem of weakly annotated data different than previous approaches (see also point 2).
>
> [1]Yin, B., Balvert, M., Zambrano, D., Schönhuth, A., & Bohte, S. (2018). An image representation based convolutional network for DNA classification. International Conference on Learning Representations (ICLR), arXiv preprint arXiv:1806.04931.
> [2] Chen, X., Liu, C., & Song, D. (2018). Towards Synthesizing Complex Programs from Input-Output Examples. International Conference on Learning Representations (ICLR), arXiv preprint arXiv:1706.01284.
> [3] Falcon, W., & Schulzrinne, H. (2018). Predicting Floor-Level for 911 Calls with Recurrent Neural Networks and Smartphone Sensor Data. International Conference on Learning Representations (ICLR), arXiv preprint arXiv:1710.11122.
>
> Ad Cons 2.)
>
> The technical challenge of this task is a difficult problem in machine learning, namely, how to deal with weakly annotated data. Instead of single instances that are labeled, a set of instances is labeled. A machine learning method has to collect and combine hints from the instances for accurate predictions. This is different from the problem setting of object recognition in MNIST, CIFAR, and ImageNet, where each image is clearly and unambiguously labeled. Current methods ameliorate this problem by learning a representation per image-patch or instance, e.g. a cell, and then pooling over those patches, such as mean- or max-pooling, or noisy-and (Kraus et al., 2016). Thus, these approaches collect hints from different patches. With GapNet-PL, we propose an alternative to these, where hints are collected by global-average pooling layers. In contrast to other methods, pooling is not done at a high-level representation nor at the prediction level but at the low-level convolutions.
> We apologize that we have not stated this clearer and we now improved the introduction section to explain the general problem setting.
>
> Ad Cons 3.)
>
> We thank the reviewer for pointing out these problems. First, the test set of Liimatainen et al. was drawn from the same basic set of images and therefore the estimate should be comparable. Under the iid assumption, the expected values of the estimates (but admittedly not the variance) should be the same. Second, we re-implemented the approach of Liimatainen et al. and now provide a performance estimate on our test data set.
> Third, we recruited two more human experts to improve the performance estimate of human experts. The performance of the human experts now ranges from an accuracy of 64% to 72%. We also tested 25 scholars, i.e. graduate and undergraduate students with life science background and specific training at this task. Thus, we can now also provide a reliable estimate for non-expert human performance. The mean performance of scholars is 51% with a range from 34% to 66%.
> We hope that these improvements would incline the reviewer to re-think his/her current evaluation.
>
> Ad Cons 4.)
>
> We want to point out that the state-of-the-art before our work was an F1 score of ~0.50 (Liimatainen et al., 2018), which we improved very significantly by proposing GapNet-PL and by comparing other architectures. We state this clearer in the introduction and conclusion section.

---

> > ### Comment · AnonReviewer1 · 2018-11-27
> > **Rating increased by 1 for the refining but the contribution is still limited**
> >
> > I’d like to increase the rating by 1 due to the highlighting of contributions and the supplemental experiments on the human performance.
> >
> > I agree to the comments of the other two reviewers, but there is still a concern of lacking significant technical contribution.
> >
> > 	- The weakly annotated data, as you described, can be a significant problem in tasks such as object detection. But such kind of weakly annotated data has rather less effect on an image classification tasks. Since you were doing the multi-label classification image-wise (absence/presence of instances) instead of predicting both the location (such as bounding box or pixel-level prediction) and the class of instances, the samples are well-labeled in this sense.
> >
> > 	- It is not convincing to claim that the global average pooling mitigates the influence of the weakly annotated problem without any theoretical analysis. The improvement on classification scores does not necessarily mean the problem is solved by GAP.
> >
> > 	- Many works have focused on the high-resolution image classification and have given a promising performance. It can be one challenge but no novel solution to this challenge was proposed in this work.
> >
> > 	- The application-oriented papers referenced above do have strong motivations for applying methods targeting specific problems or propose novel solutions. The novelty and motivation of designing the architecture in this work are expected to be stronger and more clear.

---

### Meta-Review · Area_Chair1 · 2018-12-14
**Mixed reviews, strong application paper**

**Confidence:** 3
**Recommendation:** Accept (Poster)

**Metareview:**

The reviewers all agreed that the problem application is interesting, and that there is little new methodology, but disagreed as to how that should translate into a score. The highest rating seemed to heavily weight the importance of the method to biological application, whereas the lowest rating heavily weighted the lack of technical novelty. However, because the ICLR call for papers clearly calls out applications in biology, and all reviewers agreed on its strength in that regard, and it was well-written and executed, I would recommend it for acceptance.